# Reduction Effect of Carbon Emission Trading Policy in Decreasing PM_2.5_ Concentrations in China

**DOI:** 10.3390/ijerph192316208

**Published:** 2022-12-03

**Authors:** Zhixiong Weng, Cuiyun Cheng, Yang Xie, Hao Ma

**Affiliations:** 1Institute of Circular Economy, Beijing University of Technology, Beijing 100124, China; 2Chinese Academy for Environmental Planning, Beijing 100045, China; 3School of Economics and Management, Beihang University, Beijing 100191, China; 4Laboratory for Low-Carbon Intelligent Governance, Beihang University, Beijing 100191, China; 5BGRIMM Technology Group, Beijing 100160, China

**Keywords:** carbon market, carbon trading, climate change, air pollution, difference-in-differences model

## Abstract

Carbon emissions trading is a market-based tool for solving environmental issues. This study used a difference-in-differences (DID) approach to estimate China’s carbon trading pilots to reduce PM_2.5_ concentrations. The results of this quasi-natural experiment show that the carbon trading policy effectively reduces PM_2.5_ by 2.7 μg/m^3^. We used a propensity score matching (PSM-DID) method to minimize selection bias to construct a treatment and a control group. The results show the policy effect is robust, with a PM_2.5_ concentration reduction of 2.6 μg/m^3^. Furthermore, we employed a series of robustness checks to support our findings, which notably indicate that the effect of carbon trading on reducing PM_2.5_ differs across regions over the years. The western region of China tends to be the most easily affected region, and the early years of carbon trading show slightly greater reduction effects. Our findings provide valuable policy implications for establishing and promoting carbon trading in China and other countries.

## 1. Introduction

The Carbon Emissions Trading Scheme (CETS) is market-oriented [1,2] and has become an appealing and increasingly popular tool for regulating carbon emissions [3]. Both the global regional and national carbon markets worldwide are growing rapidly. The primary mechanism behind carbon markets reaching environmental targets depends on price externalities [4], as well as on the incentive to change the behavior of enterprises and individuals [5,6].

In China, the carbon market has made significant progress and has become one of the world’s largest carbon trading systems [7]. China’s carbon trading pilot program started in Shenzhen in 2013, followed by Beijing, Tianjin, Shanghai, Chongqing, Guangdong, Hubei, and Fujian [8]. Regional carbon trading pilots have become the primary market-based environmental policy to reduce carbon emissions. By the end of 2018, the total trading volume of China’s carbon markets had reached 273 million tons, with a total turnover of approximately 782 million USD [9]. Based on the experience of regional carbon trading pilots, China launched a unified national carbon trading scheme and operated trading in July 2022 [10]. However, the current national carbon market is still in its infancy, with only the power sector involved.

Previous studies have constructed a comprehensive analytical framework to estimate the effect of the carbon market on reducing carbon emissions. Some studies applied a difference-in-differences (DID) approach to build a quasi-natural experiment by defining a treatment and a control group based on the status of adopting the carbon trading policy [11,12]. Most studies have documented that carbon trading pilots could help reduce carbon emissions. Generally, their findings imply that carbon markets encourage firms to transition toward using cleaner energies [13,14] and promote technological innovation [6,15]. The current research related to carbon trading can be summarized in the following three aspects.

The first focuses on describing the characteristics of the carbon trading policy. Some studies have reviewed China’s carbon trading market’s process, situation, challenges, and evolving processes [16,17,18]. They provide several suggestions for promoting carbon markets, such as enhancing transparency, improving liquidity, and accelerating the development of professional talent and teams in carbon trading. Some studies summarize the forms of carbon quotas allocation for specific characteristics. They found that the allocation methods will directly affect the carbon trading volumes [19,20,21].

The second important aspect is to evaluate the impact of carbon trading policy on economic effectiveness. One category of analysis focuses on testing the market efficiency of carbon trading policies. Some scholars examined the efficient market hypothesis in the European Union Emissions Trading Scheme (EU ETS) [22]. In contrast, some scholars have tested the efficiency of Chinese carbon trading pilots [23,24]. Based on measurements of market efficiency, some scholars proposed that inefficiency in some carbon trading pilots can be explained by irrational behaviors, poor information transparency, imperfect market mechanisms, and transaction costs [8,9]. Other studies focused on forecasting carbon prices. Some studies developed a novel hybrid framework for carbon price forecasting by considering the significance of multiple influencing factors [25,26].

The third category estimates the effects of reduction on carbon emissions. Some studies have concluded that carbon trading has a considerable impact on reducing carbon emissions [10,13,27] and mitigation costs [28]. However, some scholars suggest that the reduction effect of China’s carbon trading market is weak [29,30]. Since firms are the major participants in carbon trading, some studies have focused on specific sectors at the firm level to estimate the reduction effect [11,31]. Using firm-level data, scholars assessed the effects of carbon trading on carbon emissions using a DID model [5,32]. Regarding carbon neutrality, some studies have constructed indicators to investigate whether carbon trading can improve carbon reduction performance [1,8].

Compared with the significant reduction effects on carbon emissions, the elimination of air pollution has been underestimated. Relative to the direct target of reducing carbon emissions, few studies have explored the impact of the carbon market on reducing PM_2.5_ concentrations. Consequently, the co-benefits of the carbon market are often underestimated. Thus, the effect on air quality should be an integral part of the carbon market. Heavy polluters tend to decrease the direct use of costly, unacceptably polluting energies in the trading process. Furthermore, for firms with lower abatement costs, carbon trading can boost the application of green technologies because the transformation can create considerable benefits.

In this context, our study constructed a quasi-natural experiment to estimate the effect of carbon trading on PM_2.5_, and the main contributions of our study are summarized as follows: first, this study provided a new perspective for evaluating the effectiveness of the carbon trading scheme rather than emissions. Previous studies have focused on testing the economic and environmental effectiveness of China’s carbon trading policy, and their conclusions have indicated that the carbon trading policy can significantly reduce carbon emissions. However, few studies have investigated the possibility of the carbon trading policy in reducing PM_2.5_ concentrations. Second, we used a comparative exogenous approach to identify the impacts of the carbon trading policy. Considering that only some specific cities carried out the carbon trading policy, we employed the DID and propensity score matching DID models to construct a quasi-natural experiment between the treatment and control groups. Such methods can mitigate the endogenous effects of the models and more accurately identify the policy effect. Finally, based on a wide range of city-level historical data from 2007 to 2018, we used yearly PM_2.5_ concentration data and socioeconomic and meteorological factors to comprehensively examine the policy effect. Our study provides valuable implications for strengthening policy co-benefits and considering policy differences across different regions.

The remainder of this paper is organized as follows: Section 2 describes the methodology and data, Section 3 presents the empirical results and discussions, Section 4 presents our conclusions.

## 2. Methodology and Data

### 2.1. Model Specifications

We employed a standard DID model to estimate the effect of the carbon trading policy on PM_2.5_. The DID model has been widely used in econometric analyses and is an efficient approach for assessing the impact of a specific policy. The basic specification of the DID model was to set the treatment and control groups in a quasi-natural experiment. Rather than other characteristics, the policy is the primary intervention that causes differences between the treatment and control groups. The traditional ordinary least squares (OLS) are prone to endogeneity problems, resulting in biased estimation. Even using instrumental variables to reduce the potential negative impact, finding a suitable instrumental variable is difficult. Although the DID model has obvious advantages in identifying the causal effect, it is sometimes uneasy to find a similar control group since the only difference between the treated and control groups should be exposed to the policy intervention. Compared with other methods, the advantage of the DID model is more evident and it can estimate the effect of a specific intervention or treatment by comparing changes in outcomes over time [33]. Therefore, we set the DID specifications as follows:(1)$$Y_{it}=\alpha_{0}+\beta ctspost_{it}+\theta X_{it}+\mu_{i}+\nu_{t}+\varepsilon_{it}$$
where i is the prefecture-level city and includes both those implementing and those not implementing the carbon trading policy in China; t is the year variable ranging from 2007 to 2018; $Y_{it}$ is the dependent variable of the PM_2.5_ concentration; ctspost is an interaction term that multiplies $cts_{i}$ with $post_{t}$, where $cts_{i}$ is an indicator variable that implies whether or not a city implements the carbon trading policy; $post_{t}$ is a year dummy variable indicating that the $post_{t}$ will be 1 during the carbon trading policy period, otherwise it will be 0; lastly, $X_{it}$ is a series of control variables that may affect the PM_2.5_ concentration, including the population, proportion of secondary industrial added values, share of green land areas, number of industrial firms, per capita GDP, foreign direct investment, annual shine hours, annual average temperature, and annual precipitation. Therefore, the coefficient β reports the policy effect of carbon trading on PM_2.5_. Specifically, if β is less than 0, the carbon trading policy can reduce PM_2.5_; otherwise, it increases PM_2.5_. Furthermore, we use the city-specific fixed effect $\mu_{i}$ to control for all unobservable time-invariant determinants of PM_2.5_ across cities, while we include the year-fixed effect $\nu_{t}$ to control for city trends in PM_2.5_ during different periods. In addition, $\alpha_{0}$ is a constant term and $\varepsilon_{it}$ is the error term. Standard errors are clustered at the city level to control for the serial correlation of PM_2.5_ concentration.

### 2.2. Data Sources

Table 1 summarizes the statistics of the variables between the treatment and control groups. For the dependent variable, we found that the average concentration of PM_2.5_ in the treatment group was slightly lower than in the control group. With respect to control variables, there is evidence that the treatment and control groups differ in several dimensions. For example, the population is 5.72 million in the treatment group, while it is 4.21 million in the control group. The urban green land area percentage is 39.68% in the treatment group and 38.44% in the control group. In terms of the number of industrial firms and per capita GDP, they are 2291 and 53,430 CNY, respectively, in the treatment group, while the values in the control group are 1151 and 44,930, respectively, in the control group. The foreign direct investment in the treatment group (2.09 billion dollars) is higher than in the control group (0.62 billion dollars). Moreover, the annual shine hours, average temperature, and precipitation are higher in the treatment group than in the control group. For example, these three indicators are 1772 h, 19 °C, and 444 mm.

Concerning the seven carbon trading pilots in the treatment group, there are differences in some dimensions. Beijing, Tianjin, Shanghai, and Chongqing are the municipalities that are directly under the central government, which enjoys greater political advantages. The economic development levels among carbon trading cities are different, for example, the per capita GDP in Shenzhen (189,568 CNY), Guangzhou (155,491 CNY), Beijing (150,962 CNY), Shanghai (145,767 CNY), Chongqing (68,460 CNY) are higher than other cities in 2018. Correspondingly, these big cities have larger populations than other cities. For example, the populations were 21.54 million in Beijing and 24.75 million in Shanghai in 2018, while 11.08 million in Wuhan and 10.43 million in Dongguan. To eliminate the impacts of these differences, we treated these social-economic factors as control variables in the model.

The PM_2.5_ data were collected from the China National Urban Air Quality Real-time Publishing Platform. The datasets of the dependent variables (pop, secind, greland, indfirm, perGDP, fdi) were obtained from the City Statistical Yearbook. Furthermore, we collected data on sunshine, average temperature, and precipitation from the Public Weather Forecast “2345” Platform. 

## 3. Results and Discussions

### 3.1. Effect of Reducing PM_2.5_

Table 2 presents the formal tests by reporting the coefficients estimated from the DID model. In Panel A, column (1) shows the estimated effect of implementing the carbon trading policy with a series of control variables without fixed effects. PM_2.5_ declined by 5.3 μg/m^3^, indicating that the carbon trading policy can effectively reduce the PM_2.5_ concentrations. Column (2) reports that the PM_2.5_ declines are statistically significant by 6.2 μg/m^3^ when a series of controls and city-fixed effects are included. Furthermore, the regressions in column (3) confirm that the carbon trading policy is strongly associated with reducing PM_2.5_. However, the year-fixed effect in column (3) indicates that after controlling for unobservable factors over the years, the estimated coefficient of the carbon trading policy can reduce PM_2.5_ by 2.7 μg/m^3^.

### 3.2. Parallel Trend Test

A parallel trend is a prerequisite for the reliability of the DID strategy. The basic assumption of the parallel trend is that there were no systematic differences between the treatment and control groups before the policy. We plotted the PM_2.5_ concentrations before and after implementing the carbon trading policy. The dashed vertical line in Figure 1 is the year when the carbon trading policy started, and we found that PM_2.5_ in the treatment group tended to be higher than that in the control group in most years before 2013; however, PM_2.5_ in the treatment group showed a decreasing trend after policy implementation. Therefore, Figure 1 provides evidence that the parallel trend assumption is satisfied in the quasi-natural experiment.

### 3.3. Placebo Test

Our concern with the DID model framework is that other potential changes could drive the PM_2.5_ concentration. We created a placebo test by randomly selecting a sample from the treatment group. We then estimated a regression that included city and year-fixed effects. We repeated the exercise 1000 times and obtained the coefficients of the impact of the carbon trading policy on PM_2.5_. Figure 2 shows the estimates of the placebo test, and the dashed vertical line represents the actual coefficient obtained by using the actual treatment samples. We find that the true coefficient is far from the random regression results. The falsification exercise suggests that other potential changes did not drive our results.

### 3.4. Robustness Test

#### 3.4.1. PSM-DID

##### Balance Test

For econometric analysis, we divided the factors that affect the PM_2.5_ concentrations into observable and unobservable characteristics. Although we included all possible observable factors in our DID model, some unobservable factors may still lead to bias in the estimated coefficients. The propensity score matching (PSM) approach is an efficient method for reducing endogeneity. First, we selected a series of covariates and used a logit model to estimate the propensity score for each observation. Second, we applied the calliper nearest neighbor matching approach to match each observation in the treatment group with observations in the control group based on the propensity scores. Finally, we estimated the coefficients of carbon trading on PM_2.5_ using the matched samples.

Before using the matched samples, we employed a balance test to evaluate the appropriateness of the treatment and control groups. Three approaches were used. The first measures the deviation of the normalized means for the covariates in the treatment and control groups. Appendix A shows the standardized bias across covariates between the matched and unmatched observations. We found that the standardized bias of all covariates was less than 10% and negligible in the matched sample, suggesting that the matching process was effective. We then compared the common support to ensure sufficient overlap in the characteristics of the treatment and untreated units. The results in Appendix A show that most of the observations in the treatment and untreated groups fall in the area of common support. Furthermore, we plotted a kernel density figure to test the differences between the matched treatment and control groups. The results suggest that the curve of the treatment group becomes close to the control group after matching (Figure 3). In conclusion, all three methods implied that the treatment and control groups were appropriately matched.

##### Results of the PSM-DID Regression

We obtained the results of the PSM-DID regression. Table 2 (Panel B) provides the estimated coefficients of carbon trading on PM_2.5_ with different matching ratios. In Panel B, column (1) shows that when matching one treated observation with two untreated observations, PM_2.5_ would reduce by 2.6 μg/m^3^, comparable to the estimates in Panel A. In contrast, Panel B in Table 2 reports that the results were robust when we estimated the matching ratio of one to three and one to five.

#### 3.4.2. Logarithmic Form of PM_2.5_

We provided the logarithmic form of PM_2.5_ to test the robustness of the estimation. To maintain the consistency of all variables, we adopted the logarithmic form of all positive variables. As shown in Appendix A, column (1) reports that the carbon trading policy led to a 9.5% decrease in PM_2.5_. The results obtained by running the logarithmic form regression are consistent with those in the level form, further confirming the robustness of our estimations.

#### 3.4.3. Winsorization Panels

Extreme values in the panel may bias the results and lead to incorrect estimations. Winsorization is an effective way to increase the credibility of statistical inferences. We winsorized the extreme values of each variable before estimating the DID model by replacing extreme values at two-sided thresholds. The thresholds were defined as the top 95% and bottom 5% data points. Column (2) in Appendix A shows that PM_2.5_ declined by 3.1 μg/m^3^ after implementing the carbon trading policy. The results estimated were very close to the estimates in column (3) of Table 2, implying that extreme values did not drive our results.

### 3.5. Heterogeneity Test

#### 3.5.1. Heterogeneous Effects across Regions

To address the potential differences across regions, we divided the sample into groups based on geographical and economic characteristics, including the eastern, middle, and western regions. We interacted with the region dummy variable (Eastregion, Middleregion, and Westregion) with the carbon trading policy variable (ctspost). We found that the western region was the most easily affected by the carbon trading policy, with a PM_2.5_ reduction of 8.5 μg/m^3^ when controlling for the city and year-fixed effects in Panel A in Table 3. The middle region displayed a moderate impact caused by carbon trading, with a PM_2.5_ decrease of 5.0 μg/m^3^. In contrast, the coefficient for the eastern region was also statistically significant. Nevertheless, the reduction magnitude was smaller, with a 1.3 μg/m^3^ reduction in PM_2.5_.

#### 3.5.2. Heterogeneous Effects over Years

To investigate yearly differences, we interacted with the variables of the year with carbon trading in the DID model. Panel B in Table 3 reports the coefficients modeled for the yearly heterogeneous effects. We found that the carbon trading policy could reduce PM_2.5_ annually and showed slightly greater reduction effects on PM_2.5_ in the first three years. Results indicate that when controlling for the city and year-fixed effects, the PM_2.5_ decreased by 2.2 μg/m^3^, 2.9 μg/m^3^, and 3.6 μg/m^3^ in 2013, 2014, and 2015, respectively. In contrast, the carbon trading policy reduced PM_2.5_ over the subsequent years, decreasing by 2.0 μg/m^3^, 2.1 μg/m^3^, and 3.1 μg/m^3^ in 2016, 2017, and 2018, respectively, when the city and year fixed effects were controlled. Our estimations imply that the overall reduction effects are stable, although they vary slightly over time.

### 3.6. Discussion

Numerous studies have demonstrated that the carbon market plays an important role in reducing carbon emissions in many countries. However, existing studies have mainly focused on the impact of the carbon market on reducing carbon emissions while ignoring the implications for air pollution control. In this context, this study investigated the potential impact of the carbon market on air quality. We adopted a DID approach to test the environmental effects of seven Chinese carbon trading pilots. 

Our study indicates that the carbon trading policy in China not only succeeds in promoting carbon mitigation but can also significantly reduce PM_2.5_ concentrations. The empirical analysis reports that when controlling for year and city-fixed effects, PM_2.5_ concentrations can be reduced by 2.7 μg/m^3^, implying that the carbon trading policy effectively improves air quality. Compared with the other relevant studies currently, our findings provide further evidence supporting the co-benefits of the carbon market. Some scholars used monthly PM_2.5_ concentrations and weather data for 297 Chinese cities from January 2005 to December 2017, and their findings suggest that China’s carbon trading scheme can reduce PM_2.5_ concentrations by 4.8% [34]. In contrast, some scholars found that China’s carbon trading pilots lowered haze concentration and SO_2_ emissions by 0.933 μg/m^3^ and 0.7452 tons, respectively [35]. Overall, our findings are highly consistent with these studies but differ slightly in the magnitude of the reduction, mainly due to differences in research samples, periods, and empirical design. 

Compared to other air pollution control policies, the carbon trading policy can also effectively reduce air pollutants. The Chinese government has adopted a series of policies to combat severe air pollution. For example, scholars have investigated the effectiveness of the clean winter heating policy in China [33,36]. Some scholars estimated that the net treatment effect of the clean winter heating policy would alleviate 3.4 μg/m^3^ of PM_2.5_ concentrations, the reduction effect of which is close to the carbon trading observed in this study [37]. Unlike cleaner heating policies [38], the carbon trading policy aims to promote energy transformation and green technology innovation to achieve reduction targets [39,40]. In our study, we used a sample that covers the year from 2007 to 2018 to estimate the effect of the carbon trading policy on reducing PM_2.5_. We chose five to six years before and after the carbon trading implementation year (2013) to provide a more comprehensive comparison. For the carbon trading pilots, the market-oriented policy has never been used before the carbon trading policy, and the environmental regulations pushed by the local governments were regarded as the primary policy in improving environmental quality. Therefore, implementing the carbon trading policy drives some enterprises to reduce their marginal costs of carbon emission mitigation through green technological innovation. 

Unlike coercive government regulatory policies, carbon trading is a market-based system that uses a price mechanism to reduce carbon emissions. The carbon trading price reflects firms’ marginal cost of emissions [41], which encourages carbon trading among firms with different emission reduction costs. Specifically, if a firm has a low emission reduction cost and surplus carbon quota, it can trade with firms with high costs and an insufficient carbon quota in the market. This trading helps firms meet their emission reduction targets at the lowest cost. Therefore, China’s carbon trading pilots and the ongoing national carbon market are expected to play an expanding role in improving air quality.

## 4. Conclusions

This study investigated the environmental effects of China’s carbon trading pilots on PM_2.5_ concentrations. We constructed a quasi-natural experiment based on the adoption status of the carbon trading policy. In the empirical design, we defined cities implementing carbon trading as a treatment and others as a control group and tested air pollution reduction effects using a DID model. Our estimations indicate that the carbon trading policy effectively reduced PM_2.5_ concentrations by 2.7 μg/m^3^ after controlling for a series of years and city-fixed effects. The parallel trend test, placebo check, PSM-DID estimate, and other robustness tests support our findings. Furthermore, we examined the heterogeneity of carbon trading policies. Heterogeneous tests imply that carbon trading on PM_2.5_ concentrations differs across regions over the years. Specifically, the western region of China tends to be the most easily affected, and the early years of carbon trading show slightly increasing reduction effects.

Our findings provide insightful policy value for China and other countries. The carbon trading policy has proven efficient in reducing carbon emissions and air pollution. However, the role of carbon markets in reducing air pollutants has long been underestimated. Because developing countries face the dual task of improving air quality and reducing carbon emissions, carbon markets can be a crucial means of achieving coordinated effects. Therefore, we suggest that more countries explore establishing a carbon trading system. Specifically, governments should form reasonable carbon prices to enhance the vitality of carbon markets and establish a normative supervision system to guide the market. Considering the unbalanced development, China’s carbon quota should be inclined to the less developed regions to support their development. For heavy pollution industries, we suggest governments use the measures of carbon quota and price to restrict their development. As China’s national carbon market accelerates, the Chinese government should ensure that the carbon trading pilots are aligned with the national carbon market to achieve the long-term targets of air pollution control.

Further studies should focus on revealing the micro-mechanisms of the carbon market. If firm-level carbon trading data are available in the future, we hope to identify the behavioral characteristics of the firms involved in carbon trading. In addition to examining firms’ responses to the carbon market, we would also aim to examine the impact of the carbon price and trading scale on air pollution control. Furthermore, we plan to explore the impact of the national carbon market on collaborative mitigation when data are available and provide more targeted policy suggestions.

## Figures and Tables

**Figure 1 ijerph-19-16208-f001:**
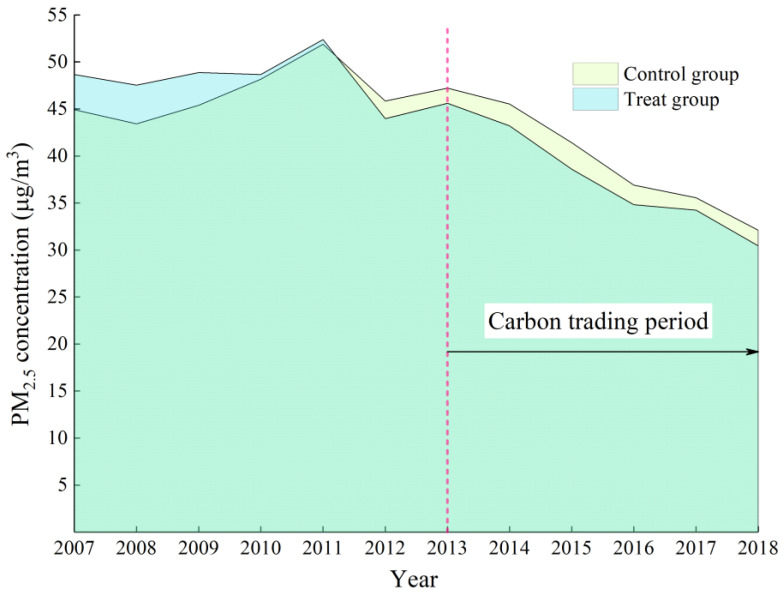
PM_2.5_ concentrations before and after the initiation of carbon trading.

**Figure 2 ijerph-19-16208-f002:**
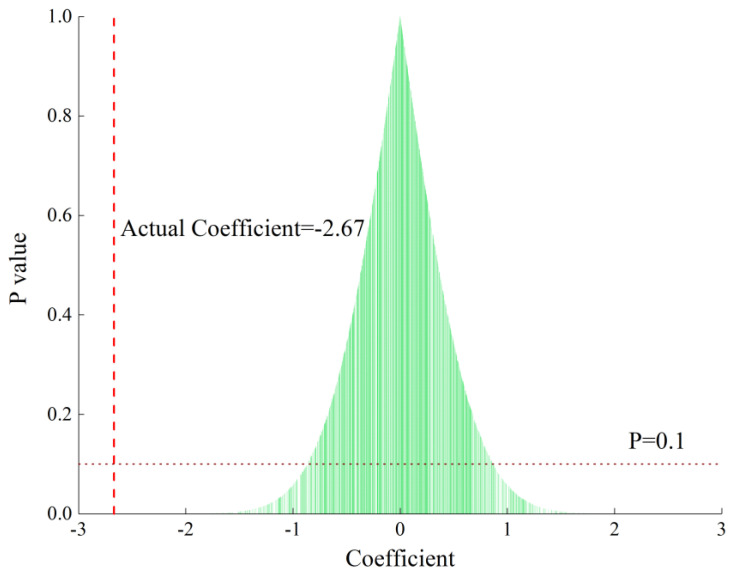
Placebo test for randomly selecting the samples.

**Figure 3 ijerph-19-16208-f003:**
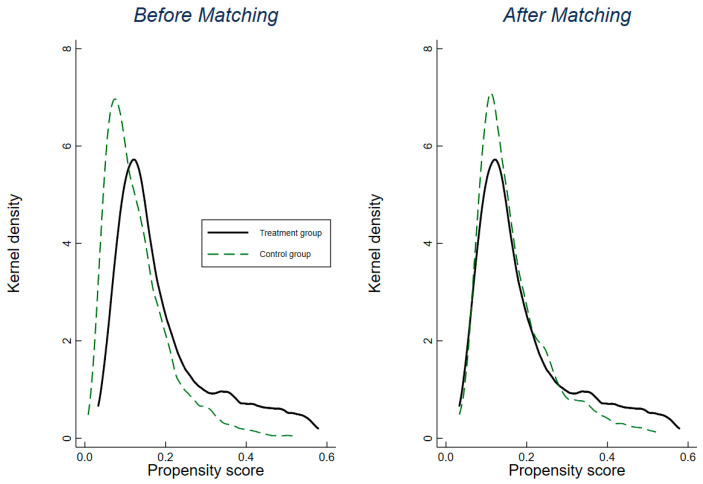
Kernel density distribution before and after matching between treat and control groups. Notes: The logarithmic forms of the covariates were used when plotting.

**Table 1 ijerph-19-16208-t001:** Summary statistics.

Variable	Meanings	Treatment			Control			Unit
		N	Mean	SD	N	Mean	SD	
pm25	PM_2.5_	444	43.1	15.4	2952	43.4	19.8	μg/m^3^
ctspost	Carbon trading	444	0.50	0.50	2952	0.00	0.00	-
pop	Population	444	5.72	551.13	2952	4.21	255.04	Million
secind	Proportion of secondary industrial added values	444	47.88	9.13	2952	48.69	11.39	%
greland	Share of green land areas	444	39.68	8.09	2952	38.44	7.81	%
indfirm	Number of industrial firms	444	2291	2647	2946	1151	1477	Number
perGDP	Per capita GDP	444	53,430	44,121	2933	44,930	121,259	CNY
fdi	Foreign direct investment	444	2.09	4.15	2952	0.62	1.27	Bill. dollar
totshine	Annual shine hours	444	1772	209	2952	2015	415	Hours
avgtemp	Annual average Temperature	444	19	3	2952	14	4	°C
precip	Annual precipitation	444	14,693	4266	2952	9624	4339	mm

Notes: Here, “N” represents the number of observations and “SD” indicates the standard deviation.

**Table 2 ijerph-19-16208-t002:** The effect of carbon trading on the reduction of PM_2.5_ concentrations.

**Panel A**	**(1)** **PM_2.5_**	**(2)** **PM_2.5_**	**(3)** **PM_2.5_**
ctspost	−5.3 ***	−6.2 ***	−2.7 ***
	(1.2)	(0.8)	(0.6)
N	3400	3400	3400
Adjusted R square	0.3588	0.8777	0.9303
**Panel B**	**Matching ratio 1:2** **PM_2.5_**	**Matching ratio 1:3** **PM_2.5_**	**Matching ratio 1:5** **PM_2.5_**
ctspost	−2.6 ***	−2.6 ***	−2.7 ***
	(0.9)	(0.8)	(0.7)
N	1086	1335	1727
Adjusted R square	0.9411	0.9429	0.9401
Controls	YES	YES	YES
City fixed-effect	NO	YES	YES
Year fixed-effect	NO	NO	YES

Notes: Standard errors are clustered at the city level with robust standard errors in parentheses. “N” represents the number of observations. Significance levels: *** *p* < 0.01.

**Table 3 ijerph-19-16208-t003:** Heterogeneous effect of carbon trading across regions.

**Panel A**	**(1)** **PM_2.5_**	**(2)** **PM_2.5_**	**(3)** **PM_2.5_**
Eastregion × ctspost	−9.1 ***	−3.8 ***	−1.3 *
	(1.5)	(0.8)	(0.7)
Middleregion × ctspost	4.5 **	−10.8 ***	−5.0 ***
	(1.8)	(0.8)	(0.8)
Westregion × ctspost	−74.6 ***	−8.4 ***	−8.5 ***
	(6.8)	(1.2)	(0.9)
N	3400	3400	3400
Adjusted R square	0.3836	0.8786	0.9306
**Panel B**	**(4)** **PM_2.5_**	**(5)** **PM_2.5_**	**(6)** **PM_2.5_**
2013 × ctspost	5.0 *	1.4 **	−2.2 ***
	(2.6)	(0.7)	(0.8)
2014 × ctspost	−2.6	−3.3 ***	−2.9 ***
	(2.6)	(0.7)	(0.6)
2015 × ctspost	−5.0 *	−5.4 ***	−3.6 ***
	(2.6)	(0.9)	(0.8)
2016 × ctspost	−3.1	−7.3 ***	−2.0 **
	(2.7)	(1.1)	(0.9)
2017 × ctspost	−10.6 ***	−9.7 ***	−2.1 **
	(2.6)	(1.0)	(0.9)
2018 × ctspost	−14.9 ***	−13.4 ***	−3.1 ***
	(2.6)	(1.3)	(1.1)
N	3400	3400	3400
Adjusted R square	0.3647	0.8816	0.9302
Controls	YES	YES	YES
City fixed-effect	NO	YES	YES
Year fixed-effect	NO	NO	YES

Notes: The “Eastregion”, “Middleregion”, and “Westregion” are the dummy variables representing the eastern, middle, and western regions in China. “N” represents the number of observations. Standard errors are clustered at the city level, with robust standard errors in parentheses. Significance levels: *** *p* < 0.01, ** *p* < 0.05, * *p* < 0.1.

## Data Availability

Not applicable.

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
