# Peer review of "Reduction Effect of Carbon Emission Trading Policy in Decreasing PM2.5 Concentrations in China"

_ijerph, 2022, doi:10.3390/ijerph192316208_

Round 1

Reviewer 1 Report

The present work treats about air pollution reduction effect of carbon emissions trading policy in China. Personally, I do not like this manuscript since I do not find it interesting nor novelty. Nevertheless, if the editor considers that it can be publishable, I suggest the following modifications:

- Highlight the novelty and interest for the scientific community in the introduction section.

- Join sections 1 and 2 into a single section called “Introduction”.

- Join sections 4 and 5 into a single section called “Results and Discussion”.

- Re-name section 6 as “Conclusions”.

- Indicate what N and SD mean in Table 1.

- Lines 207, 211 and 364 mention an appendix that does not exists. Idem regarding figures A1 and A2.

- Line 219, do not use “we”.

- Review the style of the references section according to the rules of the journal.

Author Response

Response to Reviewer #1

GENERAL COMMENTS:

The present work treats about air pollution reduction effect of the carbon emissions trading policy in China. Personally, I do not like this manuscript since I do not find it interesting or a novelty. Nevertheless, if the editor considers that it can be publishable, I suggest the following modifications:

[Response] Thanks for your valuable comments. Point-by-point revisions were provided as follows:

SPECIFIC COMMENTS:

  1. Highlight the novelty and interest of the scientific community in the introduction section.

[Response] Thanks for your valuable comments. The novelty and interest of the scientific community in the introduction section have been revised as follows:

In this context, our study constructed a quasi-natural experiment to estimate the effect of carbon trading on PM2.5, and the main contributions of our study are summarized as follows: First, this study provided a new perspective for evaluating the effectiveness of the carbon trading scheme rather than emissions. Previous studies focused on testing the economic and environmental effectiveness of China’s carbon trading policy, and their conclusions indicate that the carbon trading policy can significantly reduce carbon emissions. However, few studies have investigated the possibility of the carbon trading policy in reducing PM2.5 concentrations. Second, we used a comparative exogenous approach to identify the impacts of the carbon trading policy. Considering that only some specific cities carried out the carbon trading policy, we employed the DID and propensity score matching DID models to construct a quasi-natural experiment between the treatment and control groups. Such methods can mitigate the endogenous effects of the models and more accurately identify the policy effect. Finally, based on a wide range of city-level historical data from 2007 to 2018, we used yearly PM2.5 concentration data, and socioeconomic and meteorological factors to comprehensively examine the policy effect. Our study provides valuable implications for strengthening policy co-benefits and considering policy differences across different regions.

  1. Join sections 1 and 2 into a single section called “Introduction”

[Response] Thanks for your valuable comments. Sections 1 and 2 have now been combined as a new Introduction.

  1. Join sections 4 and 5 into a single section called “Results and Discussion”

[Response] Thanks for your valuable comments. Sections 4 and 5 have now been combined as a new Results and Discussion section.

  1. Re-name section 6 as “Conclusions”

[Response] Thanks for your valuable comments. We renamed it Conclusion.

  1. Indicate what N and SD mean in Table 1

[Response] Thanks for your valuable comments. We clarify that “N” represents the number of observations and “SD” indicates the standard deviation in Table 1 and other tables.

  1. Lines 207, 211, and 364 mention an appendix that does not exist. Idem regarding figures A1 and A2

[Response] Thanks for your valuable comments. We provided figures 1 and 2 in the Appendix, please check the Appendix as follows:

  1. Line 219, do not use “we”

[Response] Thanks for your valuable comments. Line 219 has been revised.

  1. Review the style of the references section according to the rules of the journal.

[Response] Thanks for your valuable comments. The style of the references has been revised according to the rules of the Journal. 

Reviewer 2 Report

The paper is focus on to estimating the impact of the China’s carbon emission trading on PM2.5 reduction. The authors mainly use difference-in-differences (DID) approach to analyse the impact of China’s carbon emission trading pilots.

Comments:

a.      General comment

1.      A study investigating effect of carbon emission trading to the actual air pollution reduction is still scarce. The finding of such studies can be used as insight for policy maker and governments to encourage companies to engage in carbon trading initiatives. Therefore, by its rarity alone, this paper already gives a noteworthy contribution to the body of knowledge in the field. Furthermore, the paper incorporates various reliability check such as placebo test and balance test, PSM-DID, logarithmic format test, and heterogeneity test which makes the result more credible and robust.

2.      Despite using a generic phrase “air pollution” in the title, the author of this paper is surprisingly defining air pollution as only PM2.5 concentrations. The term air pollution in the title can be perceived as overgeneralisation that potentially mislead the readers.

3.      Although, the authors state in several places (i.e., in the line 48-50 and line 289-291) about the rarity of studies on reduction of air pollution, there will be a confusion for readers on the focus of the paper because the reduction of PM2.5 which is analysed in this study is not only the result of “elimination of air pollution” but also the effect of reducing carbon emission. Please clarify the term being used to avoid confusion. 

b.      Comment on a specific part of the paper

4.      When discussing previous studies on the topic of characteristics of China’s carbon emission trading in general, (i.e.  in the line 76-81) the authors categorize the previous studies into only two groups, and for each group the authors cited only one paper despite using “some studies” at the beginning of each category. This disagreement between claims and provided evidence can be improved by citing more paper for each category.

5.      Again, like point number 3, When discussing previous studies on the topic of efficiency of China’s carbon emission trading in general (i.e., in the line 82-91) or the impact on air pollution (i.e., in the line 98-101) the authors cited only one paper despite using “some scholars” or “some studies” at the beginning of each category. This disagreement between claims and provided evidence can be improved by citing more paper for each category.

6.      The format of table 2, 3, 4 and 5 need to be improved properly as they seem to be directly copied from a statistical analysis application output.

7.      The information provided in table 5 and 6 seem too simple to put in a table format. If the information should be put in a table, the arrangement of the data in the table must be improved properly.

8.      There is no section elaborating on the limitation of the use of the DID methods for the case. There is no comparison between DID and other methods to evaluate impact of a policy either.

9.      Since the data are from 2007 – 2018 and carbon emission trading started being implemented in China in 2013, the discussion section can be enriched by comparing the difference between before and after carbon emission trading were initiated in some pilots in 2013 or at least the reason why prior CET data must start from 2007 must be explained.

10.   The differences between 7 pilots involved in this studies were not adequately discussed, the difference in population, urban green land area percentage, the number of industrial firms, per capita GDP, average temperature, and annual precipitation were not discussed adequately.

Author Response

Response to Reviewer #2

GENERAL COMMENTS:

The paper is focused on to estimating the impact of China’s carbon emission trading on PM2.5 reduction. The authors mainly use a difference-in-differences (DID) approach to analyze the impact of China’s carbon emission trading pilots.

A study investigating the effect of carbon emission trading on actual air pollution reduction is still scarce. The finding of such studies can be used as insight for policy maker and governments to encourage companies to engage in carbon trading initiatives. Therefore, by its rarity alone, this paper already gives a noteworthy contribution to the body of knowledge in the field. Furthermore, the paper incorporates various reliability checks such as the placebo test and balance test, PSM-DID, logarithmic format test, and heterogeneity test which makes the result more credible and robust.

[Response] Thanks for your valuable comments. Point-by-point revisions were provided as follows:

SPECIFIC COMMENTS:

  1. Despite using the generic phrase “air pollution” in the title, the author of this paper is surprisingly defining air pollution as only PM2.5 concentrations. The term air pollution in the title can be perceived as an over-generalization that potentially misleads the readers.
  2. Although, the authors state in several places (i.e., in lines 48-50 and lines 289-291) about the rarity of studies on the reduction of air pollution, there will be confusion for readers on the focus of the paper because the reduction of PM2.5 which is analyzed in this study is not only the result of “elimination of air pollution” but also the effect of reducing carbon emission. Please clarify the term being used to avoid confusion.

[Response] Thanks for your valuable comments. The title of the paper has been revised to “Reduction effect of carbon emission trading policy in decreasing PM2.5 concentrations in China”.

  1. When discussing previous studies on the topic of characteristics of China’s carbon emission trading in general, (i.e. in lines 76-81) the authors categorize the previous studies into only two groups, and for each group, the authors cited only one paper despite using “some studies” at the beginning of each category. This disagreement between claims and provided evidence can be improved by citing more papers for each category.
  2. Again, like point number 3, When discussing previous studies on the topic of the efficiency of China’s carbon emission trading in general (i.e., in lines 82-91) or the impact on air pollution (i.e., in lines 98-101) the authors cited only one paper despite using “some scholars” or “some studies” at the beginning of each category. This disagreement between claims and provided evidence can be improved by citing more papers for each category.

[Response] Thanks for your valuable comments. The literature review has been revised and more related studies have been cited. Based on another reviewer’s suggestions, we combined the Introduction and Literature review together.

  1. The format of tables 2, 3, 4, and 5 need to be improved properly as they seem to be directly copied from a statistical analysis application output.
  2. The information provided in Tables 5 and 6 seems too simple to put in a table format. If the information should be put in a table, the arrangement of the data in the table must be improved properly.

[Response] Thanks for your valuable comments. The tables have been combined into Tables 2 and 3, as well as Table A1 in the Appendix. The revisions are as follows:

Table 2. The effect of carbon trading on the reduction of PM2.5 concentrations.

Panel A

(1)

PM2.5

(2)

PM2.5

(3)

PM2.5

ctspost

-5.3***

-6.2***

-2.7***

(1.2)

(0.8)

(0.6)

N

3,400

3,400

3,400

Adjusted R square

0.3588

0.8777

0.9303

Panel B

Matching ratio 1:2

PM2.5

Matching ratio 1:3

PM2.5

Matching ratio 1:5

PM2.5

ctspost

-2.6***

-2.6***

-2.7***

(0.9)

(0.8)

(0.7)

N

1086

1335

1727

Adjusted R square

0.9411

0.9429

0.9401

Controls

YES

YES

YES

City fixed-effect

NO

YES

YES

Year fixed-effect

NO

NO

YES

Notes: Standard errors are clustered at the city level with robust standard errors in parentheses. “N” represents the number of observations. Significance levels: ***p<0.01, **p<0.05, *p<0.1.

 Table 3. Heterogeneous effect of carbon trading across regions.

Panel A

(1)

PM2.5

(2)

PM2.5

(3)

PM2.5

Eastregion*ctspost

-9.1***

-3.8***

-1.3*

(1.5)

(0.8)

(0. 7)

Middleregion*ctspost

4.5**

-10.8***

-5.0***

(1.8)

(0.8)

(0.8)

Westregion*ctspost

-74.6***

-8.4***

-8.5***

(6.8)

(1.2)

(0.9)

N

3,400

3,400

3,400

Adjusted R square

0.3836

0.8786

0.9306

Panel B

(4)

PM2.5

(5)

PM2.5

(6)

PM2.5

2013*ctspost

5.0*

1.4**

-2.2***

(2.6)

(0.7)

(0.8)

2014*ctspost

-2.6

-3.3***

-2.9***

(2.6)

(0.7)

(0.6)

2015*ctspost

-5.0*

-5.4***

-3.6***

(2.6)

(0.9)

(0.8)

2016*ctspost

-3.1

-7.3***

-2.0**

(2.7)

(1.1)

(0.9)

2017*ctspost

-10.6***

-9.7***

-2.1**

(2.6)

(1.0)

(0.9)

2018*ctspost

-14.9***

-13.4***

-3.1***

(2.6)

(1.3)

(1.1)

N

3,400

3,400

3,400

Adjusted R square

0.3647

0.8816

0.9302

Controls

YES

YES

YES

City fixed-effect

NO

YES

YES

Year fixed-effect

NO

NO

YES

Notes: The “Eastregion” “Middleregion” “Westregion” are the dummy variables representing the eastern, middle, and western regions in China. “N” represents the number of observations. Standard errors are clustered at the city level with robust standard errors in parentheses. Significance levels: ***p<0.01, **p<0.05, *p<0.1.

  1. There is no section elaborating on the limitation of the use of the DID methods for the case. There is no comparison between DID and other methods to evaluate the impact of a policy either.

[Response] Thanks for your valuable comments. The revised model descriptions have been provided in the Model specifications as follows:

We employed a standard DID model to estimate the effect of the carbon trading policy on PM2.5. The DID model has been widely used in econometric analyses and is an efficient approach for assessing the impact of a specific policy. The basic specification of the DID model was to set the treatment and control groups in a quasi-natural experiment. Rather than other characteristics, the policy is the primary intervention that causes differences between the treatment and control groups. The traditional ordinary least squares (OLS) are prone to endogeneity problems, resulting in biased estimation. Even using instrumental variables to reduce the potential negative impact, finding a suitable instrumental variable is difficult. Although the DID model has obvious advantages in identifying the causal effect, it is sometimes uneasy to find a similar control group since the only difference between the treated and control groups should be exposed to the policy intervention. Compared with other methods, the advantage of the DID model is more evident and it can estimate the effect of a specific intervention or treatment by comparing changes in outcomes over time.

  1. Since the data are from 2007 – 2018 and carbon emission trading started being implemented in China in 2013, the discussion section can be enriched by comparing the difference between before and after carbon emission trading was initiated in some pilots in 2013 or at least the reason why prior CET data must start from 2007 must be explained.

[Response] Thanks for your valuable comments. The relevant revisions were provided in the Discussion section as follows:

Compared to other air pollution control policies, the carbon trading policy can also effectively reduce air pollutants. The Chinese government has adopted a series of policies to combat severe air pollution. For example, scholars have investigated the effectiveness of the clean winter heating policy in China [36, 39]. Some scholars estimated that the net treatment effect of the clean winter heating policy would alleviate 3.4 μg/m3 of PM2.5 concentrations, the reduction effect of which is close to the carbon trading observed in this study [40]. Unlike cleaner heating policies [41], the carbon trading policy aims to promote energy transformation and green technology innovation to achieve reduction targets [42, 43]. In our study, we used a sample that covers the year from 2007 to 2018 to estimate the effect of the carbon trading policy on reducing PM2.5. We chose five to six years before and after the carbon trading implementation year (2013) to provide a more comprehensive comparison. For the carbon trading pilots, the market-oriented policy has never been used before the carbon trading policy, and the environmental regulations pushed by the local governments were regarded as the primary policy in improving environmental quality. Therefore, implementing the carbon trading policy drives some enterprises to reduce their marginal costs of carbon emission mitigation through green technological innovation.

  1. The differences between the 7 pilots involved in this study were not adequately discussed, and the difference in population, urban green land area percentage, the number of industrial firms, per capita GDP, average temperature, and annual precipitation were not discussed adequately.

Table 1 summarizes the statistics of the variables between the treatment and control groups. For the dependent variable, we found that the average concentration of PM2.5 in the treatment group was slightly lower than in the control group. With respect to control variables, there is evidence that the treatment and control groups differ in several dimensions. For example, the population is 5.72 million in the treatment group, while it is 4.21 million in the control group. The urban green land area percentage is 39.68% in the treatment group and 38.44% in the control group. In terms of the number of industrial firms and per capita GDP, they are 2,291 and 53,430 CNY, respectively, in the treatment group, while the values in the control group are 1,151 and 44,930, respectively, in the control group. The foreign direct investment in the treatment group (2.09 billion dollars) is higher than in the control group (0.62 billion dollars). Moreover, the annual shine hours, average temperature, and precipitation are higher in the treatment group than in the control group. For example, these three indicators are 1,772 hours, 19℃, and 444 mm.

Concerning the seven carbon trading pilots in the treatment group, there are differences in some dimensions. Beijing, Tianjin, Shanghai, and Chongqing are the municipalities that are directly under the central government, which enjoys greater political advantages. The economic development levels among carbon trading cities are different, for example, the per capita GDP in Shenzhen (189,568 CNY), Guangzhou (155,491 CNY), Beijing (150,962 CNY), Shanghai (145,767 CNY), Chongqing (68,460 CNY) are higher than other cities in 2018. Correspondingly, these big cities have larger populations than other cities. For example, the populations were 21.54 million in Beijing and 24.75 million in Shanghai in 2018, while 11.08 million in Wuhan and 10.43 million in Dongguan. To eliminate the impacts of these differences, we treated these social-economic factors as control variables in the model.

Round 2

Reviewer 2 Report

No further comment